# *Lactobacillus rhamnosus* CY12 Enhances Intestinal Barrier Function by Regulating Tight Junction Protein Expression, Oxidative Stress, and Inflammation Response in Lipopolysaccharide-Induced Caco-2 Cells

**DOI:** 10.3390/ijms231911162

**Published:** 2022-09-22

**Authors:** Juanshan Zheng, Anum Ali Ahmad, Yayuan Yang, Zeyi Liang, Wenxiang Shen, Min Feng, Jiahao Shen, Xianyong Lan, Xuezhi Ding

**Affiliations:** 1Key Laboratory of Yak Breeding Engineering, Lanzhou Institute of Husbandry and Pharmaceutical Sciences, Chinese Academy of Academy of Agricultural Sciences, Lanzhou 730050, China; 2Laboratory of Animal Genome and Gene Function, College of Animal Science and Technology, Northwest A&F University, Yangling, Xianyang 712100, China; 3State Key Laboratory of Grassland Agro-Ecosystems, School of Life Sciences, Lanzhou University, Lanzhou 730050, China; 4Key Laboratory of Veterinary Pharmaceutical Development, Ministry of Agricultural and Rural Affairs, Lanzhou Institute of Husbandry and Pharmaceutical Sciences, Chinese Academy of Agricultural Sciences, Lanzhou 730050, China

**Keywords:** *Lactobacillus rhamnosus*, lipopolysaccharide, oxidative stress, intestinal tight junction barrier, intestinal inflammation

## Abstract

The intestinal barrier is vital for preventing inflammatory bowel disease (IBD). The objectives of this study were to assess whether the *Lactobacillus rhamnosus* CY12 could alleviate oxidative stress, inflammation, and the disruption of tight junction (TJ) barrier functions induced by lipopolysaccharide (LPS), and therefore to explore the potential underlying molecular mechanisms. Our results showed that LPS-induced Cancer coli-2 (Caco-2) cells significantly increased the levels of reactive oxygen species (ROS), lactate dehydrogenase, inflammatory cytokines interleukin-1β, interleukin-6, interleukin-8, and tumor necrosis factor-α (IL-1β, IL-6, IL-8, and TNF-α), and the cell apoptosis rate while decreasing the levels of TJ proteins occludin, zonula occludens-1 (ZO-1), and claudin and antioxidant enzymes, such as catalase, superoxide dismutase, and glutathione peroxidase(CAT, SOD, and GSH-Px) (*p* < 0.05). However, *Lactobacillus rhamnosus* CY12 could relieve cytotoxicity, apoptosis, oxidative stress, and pro-inflammatory cytokine expressions, and also inhibit the Toll-like receptor 4/nuclear factor kappa-B(TLR4/NF-κB) signaling pathway. Furthermore, the gene expression of antioxidant enzymes, as well as the mRNA and protein expressions of TJ proteins, was improved. Particularly, the concentration of 10^8^ cfu/mL significantly prevented the inflammatory injury induced by LPS in Caco-2 cells (*p* < 0.05). These findings support a potential application of *Lactobacillus rhamnosus* CY12 as a probiotic to prevent LPS-induced intestinal injury and treat intestinal barrier dysfunction.

## 1. Introduction

Intestinal barrier function (IBF) not only refers to the capacity of the intestine to regulate the transport of nutrients, water, and electrolytes but is also an important defense line against the invasion of pathogenic bacteria from the external environment [1]. Damage to the intestinal barrier function caused by pathogens contributes to abnormal intestinal inflammation and the development of chronic inflammatory diseases such as inflammatory bowel diseases (IBD) by promoting the release of pro-inflammatory cytokines and the disruption of tight junctions (TJs) [2]. Therefore, the maintenance of the integrity of the intestinal epithelial barrier plays a crucial role in inflammatory bowel diseases, and it has become a therapeutic and preventive target for IBDs [3,4]. An increasing amount of evidence demonstrates that inflammatory diseases are associated with the overproduction of inflammatory cytokines, such as tumor necrosis factorα (TNF-α), interleukin1β (IL-1β), and interleukin 6 and 8 (IL-6 and IL-8) by NF-κB pathways [5,6]. These pro-inflammatory cytokines potentially impair IBF and increase intestinal permeability. It has been suggested that several potential mechanisms are referred to as driving the disruption of the pro-inflammatory-cytokines-induced intestinal barrier, including low expression of TJ-associated proteins [4,7]. In particular, oxidative stress seems to have great importance in intestinal barrier dysfunction, inflammation responses, and inflammatory bowel disease as it decreases the mRNA expression and distribution of TJ proteins in an epithelial cell model in vitro [8,9]. The TJ complexes and other alterations in TJ proteins have been implicated as possible key participants involved in the epithelial barrier function in IBD [10,11].

Lipopolysaccharide (LPS), a vital component of the *E. coli* outer membrane, can cause damage to the intestinal epithelial barrier and induce chronic intestinal inflammation [12,13]. LPS contributes to immune defense by interrupting intestinal TJs and inducing systemic inflammatory responses [14,15]. Consequently, LPS was widely used to establish the inflammation correlation model. Toll-like receptors (TLRs) are crucial pattern-recognition receptors of the innate immune system which are indispensable for sensing intestinal microbiota to maintain the homeostasis of intestinal epithelial cells and to protect from epithelial injury [16]. LPS is recognized by TLR4, which serves in the activation of the transcription factor NF-κB and the secretion of inflammatory cytokines, such as TNF-α, IL-6, and IL-1β [17]. Therefore, it is essential to improve the intestinal barrier by inhibiting unnecessary inflammatory responses.

As one of the most important bacterial groups in the food industry, the application of beneficial microbes, also known as ‘probiotics’, is considered to be an important strategy to maintain host health [18]. Increasing evidence has indicated that probiotics have various potential benefits for the host, such as excluding or inhibiting pathogens [19], improved digestion [20], strengthening the intestinal epithelial barrier [21], improved immunity [22], maintaining intestinal balance [23], and management of inflammatory bowel diseases [23]. Additionally, potential probiotic microorganisms and probiotics have demonstrated an astonishing antagonistic activity toward a wide range of sturdy food and clinical pathogens [24,25]. The importance of probiotics in human–animal nutrition is widely recognized [26]. It was reported that the *Bifidobacterium bifidum* (*B. bifidum*) strain could prevent the TNF-α stimulated damage of the intestinal epithelial barrier in the Caco-2 (Cancer coli-2) cells model [27]. Another study also has indicated that *bifidobacterium dentium* N8 significantly relieved LPS-induced intestinal barrier injury by upregulating the TJ proteins and alleviating the inflammatory response in the Caco-2 cells model [28]. Furthermore, previous studies have demonstrated that *Lactobacillus rhamnosus* GG plays an important role in promoting intestine immune development and protection against inflammation-stimulated damage to the intestinal barrier function [22,29]. Numerous studies have shown that *Lactobacillus rhamnosus* could reduce LPS-induced inflammation in the epithelial cells by inhibiting the NF-κB signaling pathway [27,30]. However, in recent years, a growing body of research has revealed that the functions of probiotics are highly strain-specific, and their biological effects should be individually evaluated [22]. Our previous work reported the isolation of the *Lactobacillus rhamnosus* CY12 (*L. rhamnosus* CY12) strain from cattle-yak milk that displayed potential probiotic characteristics such as a high survival rate in acidic conditions and bile salts, high antimicrobial activity, and high adhesive potential [31]. Considering the specificity of the strains, the protective effects of the novel *L. rhamnosus* CY12 strain isolated from cattle-yak milk on intestinal inflammation are worthy for further study. In the present study, we aimed to determine the effects of *L. rhamnosus* CY12 on LPS-induced oxidative stress, TJ destruction, inflammatory responses, and apoptosis in Caco-2 cells. Moreover, the potential mechanism was also explored. We hypothesized that a novel *L. rhamnosus* CY12 strain isolated from cattle-yak milk would exert inhibitory effects on damage to the intestinal barrier function induced by LPS in Caco-2 cells.

## 2. Results

### 2.1. Effect of L. rhamnosus CY12 on Cell Viability and Cytokines Expression in LPS-Induced Caco-2 Cells

The results for cell viability showed that *L. rhamnosus* CY12 had no toxic effects on Caco-2 cells when the concentrations were less than 10^10^ cfu/mL (*p* < 0.05) (Figure 1A). So, the concentrations of 10^7^, 10^8^, and 10^9^ of *L. rhamnosus* CY12 were used in the further experiments by keeping cytotoxicity and previous recommendations in mind [22].

Compared with the LPS group, downregulation of the levels of IL-1β, IL-6, and TNF-α (*p* < 0.05) was recorded in the C group and *L. rhamnosus* CY12 pretreated groups (Figure 1B–D). Moreover, the IL-1β and TNF-α levels in the C and M groups were significantly lower compared to the L and H groups (*p* < 0.05). However, no significant difference in the level of IL-6 was observed between the considered groups (*p* > 0.05; Figure 1C).

To assess the cytotoxicity effect of *L. rhamnosus* CY12 on Caco-2 cells, the release of LDH was detected using the LDH assay kit. As shown in Figure 1E, the activity of LDH in each group showed a significant difference (*p* < 0.05). The results revealed that the activity of LDH in the LPS group significantly increased compared with other groups (*p* < 0.05), while the M group showed significantly lower LDH activity (*p* < 0.05).

### 2.2. Effects of L. rhamnosus CY12 on Oxidative Stress and Apoptosis in LPS-Induced Caco-2 Cells

The activity of GSH in the C and M groups was significantly higher compared to other groups (*p* < 0.05) (Figure 1F). Compared with the LPS group, the *L. rhamnosus* CY12 significantly decreased the intracellular concentration of ROS (Figure 2), indicating *L. rhamnosus* CY12 could alleviate LPS-induced oxidative stress.

Compared to the control group, the apoptotic rate in LPS-induced cells was significantly higher (*p* < 0.05) (Figure 3). Moreover, *L. rhamnosus* CY12 remarkably reduced the cell apoptosis rate.

### 2.3. Effects of L. rhamnosus CY12 on the Expression of Genes Related to Inflammation in LPS-Induced Caco-2 Cells

Compared with the LPS group, *L. rhamnosus* CY12 groups significantly decreased the expression of genes related to inflammation (*p* < 0.05), as shown in Figure 4A–D. The expression level of IL-1β in the C and M groups was remarkably lower than that of other groups (*p* < 0.05). In comparison with the L and LPS groups, TNF-α and IL-8 mRNA levels significantly decreased in other groups (*p* < 0.05). In addition, IL-6 mRNA showed lower expression in the L and M groups compared with the H and LPS groups (*p* < 0.05). The Caco-2 cells pretreated with *L. rhamnosus* CY12 showed significant downregulation of mRNA levels of TLR4 and NF-κB in a dose-dependent manner (*p* < 0.05). The expression of NF-κB levels in the M and H groups was lower compared with the LPS group (*p* < 0.05) (Figure 4E,F). However, the mRNA level of TLR4 showed no significant difference in the L and M groups compared to the control group (*p* > 0.05).

### 2.4. Effects of L. rhamnosus CY12 on Gene Expression of TJ Proteins in LPS-Induced Caco-2 Cells

Compared to the control group, the Caco-2 cells stimulated with LPS for 4 h showed a significant downregulation of gene expression for occludin, ZO-1, and claudin (*p* < 0.05), as shown in Figure 4G–I. The gene expression of TJ proteins was noticeably higher in a dose-dependent manner in Caco-2 cells pretreated with *L. rhamnosus* CY12 (*p* < 0.05).

Compared to the L group, the mRNA level of occludin in the M group was higher (*p* < 0.05). The mRNA level of ZO-1 was significantly upregulated in the L and M groups compared to the H and LPS groups (*p* < 0.05), while the H group showed higher expression than the LPS group (*p* < 0.05). Moreover, compared with the LPS group, the claudin showed higher expression in the L, M, and H groups (*p* > 0.05).

### 2.5. Effects of L. rhamnosus CY12 on Gene Expression of Antioxidant Enzymes in LPS-Induced Caco-2 Cells

The genes related to oxidative stress showed significantly increased expression in *L. rhamnosus*-CY12-treated cells (*p* < 0.05) (Figure 4J–L). The mRNA levels of CAT and GSH-Px in the M group were significantly higher than in the L, H, and LPS groups (*p* < 0.05), while no change between the L and H groups (*p* > 0.05) was observed. The gene of SOD showed higher expression in the H group than in the L, M, and LPS groups (*p* < 0.05), and lower expression in the LPS group was observed compared with the M group (*p* < 0.05). These results indicated that *L. rhamnosus* CY12 enhanced the antioxidation and effectively inhibited oxidative stress in LPS-induced Caco-2 cells.

### 2.6. Effects of L. rhamnosus CY12 on TJ Proteins and TLR4/NF-κB p65 Protein Expression in LPS-Induced Caco-2 Cells

As shown in Figure 5, the Caco-2 cells in the LPS group showed an upregulated expression of TLR4 and p65 proteins compared to the control group (*p* < 0.05), while the expression levels of TJ proteins significantly increased in Caco-2 cells treated with *L. rhamnosus* CY12 (*p* < 0.05). Compared with the L, H, and LPS groups, the expression levels of claudin and occludin proteins were remarkably upregulated in the M groups (*p* < 0.05). The group treated with *L. rhamnosus* CY12 showed upregulation in the expression of the ZO-1 protein compared with LPS group (*p* < 0.05). No change in the L, M, and H groups (*p* > 0.05) was recorded.

Compared to the control group, LPS remarkably upregulated the expression of TLR4 and p65 (*p* < 0.05) proteins in Caco-2 cells. On the other hand, *L. rhamnosus*-CY12-treated cells showed downregulated expression of TLR4 and p65 (*p* < 0.05) proteins. Compared with the L and H group, the expression levels of TLR4 proteins decreased in the M group (*p* < 0.05), while the p65 protein displayed no significant difference in the L, M, and H groups (*p* > 0.05). The TLR4 protein showed no significant difference between the L and H groups (*p* > 0.05). These results indicated that *L. rhamnosus* CY12 downregulated the inflammatory response by inhibiting the TLR4/NF-κB signaling pathway.

We also examined the capacity of *L. rhamnosus* CY12 to recover the TJ barrier function with immunocytochemistry. Our results found that LPS decreased the abundance of TJ proteins compared to the control group. The treatment of LPS-induced Caco-2 cells with *L. rhamnosus* CY12 significantly increased the integrity of the TJ barrier function. The distribution of ZO-1 (Figure 6A) and occludin (Figure 6B) proteins in the M group was significantly improved compared to the L, H, and LPS groups, and the H group displayed better distribution compared to the L and LPS groups. Moreover, to further confirm the effects of the *L. rhamnosus* CY12 strain on the TLR4/NF-κB signaling pathway, the relative fluorescence intensity of the TLR4 (Figure 7A) and p65 (Figure 7B) was also verified with immunofluorescence. Similar to the mRNA and protein expressions, a significant decrease in the nuclear accumulation of TLR4 and p65 was observed after treatment of LPS-induced Caco-2 cells with the *L. rhamnosus* CY12 strain. These results indicated that the *L. rhamnosus* CY12 strain could inhibit TLR4 and p65 translocation.

## 3. Discussion

The intestinal epithelial cells act as a frontline defensive barrier of host mucosal immunity by tightly regulating the tight junctions against pathogenic microorganisms and potential xenobiotics [10,11,32]. The damage to the intestinal epithelial function promotes the release of pro-inflammatory cytokines, cell apoptosis, and the disruption of tight junctions (TJs), thereby contributing to increased intestinal permeability, infection, and provoked chronic inflammatory diseases [33]. Lipopolysaccharide (LPS) is a principal component of Gram-negative bacteria cell walls that is widely used in establishing in vitro inflammation models and plays a crucial role in the initiation and development of the epithelial barrier dysfunction [12,13,22]. Similar to previous LPS-induced-cell studies [13,32,34], our results found that the *L. rhamnosus* CY12 strain relieved the oxidative stress, the disruption of the intestinal TJ barrier function, and inflammatory response in LPS-induced Caco-2 cells through increasing the antioxidant enzymes (CAT, SOD, and GSH-Px) and tight junction proteins (claudin, occludin, and ZO-1) and inhibiting the pro-inflammatory cytokines and the TLR4/NF-κB signaling pathway. The regulatory mechanism induced by the *L. rhamnosus* CY12 inhibited damage to the LPS-induced intestinal barrier function in Caco-2 cells, as shown in Figure 8.

It is well known that the imbalance between enhanced levels of ROS and the lower activity of antioxidant enzymes is a crucial reason for oxidative stress, resulting in damage to DNA and protein, cell apoptosis, and gastrointestinal diseases [35,36]. Probiotics have been demonstrated to possess the remission effect on oxidative stress and cell apoptosis by upregulating the antioxidant enzymes, including CAT, SOD, and GSH-Px, as well as GSH contents [37,38,39]. Similar results were recorded in the present study indicating the potential use of the *L. rhamnosus* CY12 strain as a promising probiotic candidate.

The TJ proteins such as claudin, occludin, and ZO-1 comprise the major intestinal epithelial barrier. The damaged intestinal barrier function causes abnormal expression of TJ proteins, which allows LPS, pathogens, and other harmful substances to pass through the intestinal wall into the blood circulation and increase the activation of the immune system and the inflammatory response [14,15,40]. A previous study showed that *Lactobacillus rhamnosus Gorbach-Goldin* (LGG) played an important role in the strengthening of the mucosal immune barrier, protection against inflammation-induced damage, and maintaining intestinal homeostasis [29]. Some studies have demonstrated that probiotics prevented inflammation-induced intestinal barrier function damage by increasing the distribution and expression levels of TJ proteins [29,41,42,43]. A study reported the restoration of disturbed gut microbiota by increasing TJ proteins expression and inhibiting LPS-mediated NF-κB activation when treated with the *Lactobacillus*
*plantarum* LC27 strain [34]. In this study, we also recorded improved expression of TJ genes and proteins when Caco-2 cells were treated with the *L. rhamnosus* CY12 strain, indicating its ability to maintain intestinal barrier function.

It is generally known that LPS can stimulate the pro-inflammatory cytokines, the major representatives of which are IL-6, TNF-α, and IL-1β. The production of pro-inflammatory cytokines through the intestinal epithelial cell can cause damage to the intestinal barrier function [5,6]. Therefore, LPS is often used to build the inflammation-associated model. Reports have shown that *Lactobacillus* could prevent the inflammatory response by downregulating the expression of TNF-α, IL-1β, and IL-6 [22,44], which is in line with our findings. We assumed that effectively inhibited LPS-stimulated inflammation responses in vitro by *L. rhamnosus* CY12 might be due to the inhibition of pro-inflammatory cytokines such as IL-6, TNF-α, and IL-1β.

The transcription factor NF-κB can be activated by oxidative stressors and inflammation, and it plays an essential role in regulating many cellular processes such as immune response, inflammatory diseases, as well as cell growth and apoptosis [45,46,47]. In recent years a growing number of studies suggested that inflammatory diseases involve the overexpression of inflammatory cytokines by the NF-κB pathway, and that TLR4 is the major receptor for LPS recognition [17]. Activation of TLR4 was reported to induce the NF-κB pathway and cause the overexpression of pro-inflammatory cytokines, contributing to intestinal barrier damage and an inflammation response [33,48]. A study reported that *L. rhamnosus* JL-1 decreased the inflammation responses of LPS induced by the inhibition of the NF-κB signaling pathway [22]. Another study showed that *L. rhamnosus* GG reduced inflammation of LPS induced in epithelial cells through the NF-κB and MAPK signaling pathway [30]. In the present study, we also recorded the inhibition of the TLR4/NF-κB signaling pathway upon inoculation of the *L. rhamnosus* CY12 strain. Based on the results of this study, the action mechanism of *L. rhamnosus* CY12 initiated with the prevention of the damage of the intestinal epithelial function in LPS-induced Caco-2 cells by upregulating TJ protein expressions and reducing ROS accumulation, as well as overproducing inflammatory cytokines that activated TLR4/NF-κB signaling pathway.

## 4. Materials and Methods

### 4.1. Culture of L. rhamnosus CY12 Strain

The *L. rhamnosus* CY12 strain was cultured in de Man, Rogosa, and Sharpe agar (MRS; Guangdong huankai microbial SCI&TECH. Co., Ltd., Guangzhou, China) at 37 °C with shaking at 170 rpm overnight. The culture was then centrifuged at 6000× *g* for 10 min, and the strain was obtained in a pellet. This later was washed with sterile PBS three times, and cells were resuspended in non-supplemented MEM with different concentrations to use for in vitro experiments.

### 4.2. Cell Culture

Human colorectal adenocarcinoma cell line Caco-2 (China Cell Line Bank, Shanghai, China) was cultured in Minimum Essential Medium (MEM) supplemented with 10% (vol/vol) heat-inactivated fetal bovine serum (FBS), 1% MEM Non-Essential Amino Acids (NEAA), 1% Sodium Pyruvate (SP), and 1% L-glutamine (all reagents were from Gibco, Thermo Fisher Scientific, Shanghai, China). The Caco-2 cells were incubated in a humidified incubator with 5% CO_2_ at 37 °C for all experiments.

### 4.3. Cell Viability Assay

To detect the influence of bacteria on cell viability, Caco-2 cells were cultured into 96-well plates. When the density of cells reached 60–80%, the Caco-2 cells were treated with and without different bacteria concentrations (10^6^, 10^7^, 10^8^, 10^9^, and 10^10^ cfu/mL). After incubation for 24 h, the cell viability was evaluated using a Cell Counting Kit 8 (CCK-8) (Biosharp, Shanghai, China) according to the manufacturer’s instructions. Briefly, the cells were washed twice with sterile PBS, and 10 µL of CCK-8 solution was added to each 100 µL of cell medium. After incubation for 2–4 h in a humidified incubator, the absorbance was measured with the enzyme-labeling measuring instrument at 450 nm (OD_450_ nm), which is directly proportional to the viability of Caco-2 cells.

### 4.4. Enzyme-Linked Immunosorbent Assay (ELISA)

The Caco-2 cell line was cultured in 96-well and 6-well plates. The Caco-2 cells were treated with different bacterial (10^7^, 10^8^, and 10^9^ cfu/mL) cultures for 4 h. After 4 h, the bacterial suspension was discarded, and cells were treated with and without 1 μg/mL LPS from *Escherichia coli* 0111:B4 (Sigma, St. Louis, MO, USA) for 4 h according to a previous study, with slight modifications [22]. The Caco-2 cells were treated with the following: (1) the control group (C): without *L. rhamnosus* CY12 or LPS for negative control group; (2) Low-dose group (L): treated with 10^7^ cfu/mL *L. rhamnosus* CY12 and 1 μg/mL LPS; (3) Middle-dose group (M): treated with 10^8^ cfu/mL *L. rhamnosus* CY12 and 1 μg/mL LPS; (4) High-dose group (H): treated with 10^9^ cfu/mL *L. rhamnosus* CY12 and 1 μg/mL LPS; (5) LPS group (LPS): 1 μg/mL LPS induced Caco-2 cells for positive control group. The supernatants were harvested from Caco-2 cells after 4 h, and the levels of cytokines IL-1β, IL-6, and TNF-α were measured using the corresponding ELISA kits according to the manufacturer’s instructions (Meimian, Yancheng, China).

### 4.5. Lactate Dehydrogenase (LDH) Activity Assay

To evaluate the LDH activity, the Caco-2 cell line was cultured in 96-well plates and treated correspondingly. The supernatants were obtained in eppendorf tubes and centrifuged at 12,000 rpm for 20 min at 4 °C. The LDH assay kit (Nanjing Jiancheng Bioengineering Institute, Nanjing, China) was used to detect LDH activity according to the manufacturer’s instructions. Nicotinamide adenine dinucleotide was reduced by 1 oxidized μmol per minute and was regarded as one unit of enzyme activity. Each treatment had three independent replicates.

### 4.6. Glutathione (GSH) Activity Assay

The 96-well plates were used to culture Caco-2 cells with corresponding treatments. Cells were obtained, washed twice with sterile PBS, and sonicated in ice-cold PBS. Then, 0.1 mL precipitant was added to 0.1 mL lysates and centrifuged at 3500 rpm for 10 min. Lastly, the GSH activity in Caco-2 cells was detected according to the instructions of the GSH assay kit (Nanjing Jiancheng Bioengineering Institute, Nanjing, China).

### 4.7. Assessment of Intracellular Reactive Oxygen Species (ROS)

The ROS levels were measured by ROS assay kit as described previously [36]. Briefly, the 12-well culture plates were used to culture Caco-2 cells after the corresponding treatments. The culture medium was discarded, and cells were washed with sterile PBS. Then, cells were treated with DCFH-DA in the dark. Lastly, a laser scanning confocal microscope (ZEISS, LSM800, Oberkochen, Germany) was used to scan 12-well plates. Intracellular ROS showed green fluorescent light in the microscope.

### 4.8. Apoptosis Assay

To assess apoptosis, 12-well culture plates were used to culture Caco-2 cells with the corresponding treatments. The apoptosis was assessed using Annexin V/PI apoptosis kits (Yeasen, Shanghai, China). Briefly, the culture medium was discarded, cells were washed with sterile PBS, 500 µL of 1× binding buffer (2 µL of Annexin V-FITC and 5 µL of PI) was added to each well uniformly in the dark, and they were incubated at room temperature for 15 min. Then, cells were washed with 1× binding buffer, and a laser scanning confocal microscope (ZEISS, LSM800, Oberkochen, Germany) was used to scan plates. The early apoptotic cells showed green fluorescent light under the microscope. The late apoptotic cells showed red and green fluorescent light under the microscope, and normal cells displayed no fluorescence.

### 4.9. Assessment of mRNA Expression Levels

The 6-well plates were used to culture Caco-2 cells to evaluate mRNA expression levels. After the corresponding treatments, the total cellular RNA was extracted by a total extraction reagent (Vazyme, Nanjing, China) following the manufacturer’s protocol. The Agilent 2100 Bioanalyzer was used to detect RNA integrity (Agilent Technologies, Santa Clara, CA, USA). Then, the cDNA was synthesized from isolated total RNA by the cDNA synthesis kit (Accurate Biology, Changsha, China). PCR amplification and Real-time PCR analysis of gene expression were performed using a fluorescence thermal cycler (CFX96, Bio-Rad, Shanghai, China) according to SYBR green premix pro taq HS qPCR kit (Accurate biology, Changsha, China). The $2^{-\Delta \Delta CT}$ method was used to determine the relative expression of each candidate gene [49], while GAPDH (glyceraldehyde-3-phosphate dehydrogenase) was used for normalization. The PCR primers for tight junction, oxidative stress, and inflammation are shown in Appendix A.

### 4.10. Protein Expression Analysis

The 6-well plates were used to culture Caco-2 cells to evaluate protein expression levels. After the corresponding treatments, the capillary Western blot (Wes, Protein Simple; San Jose, CA, USA) was used to evaluate the expression levels of the ZO-1, Occludin, Claudin, TLR4, and p65. All experimental steps were performed according to the manufacturer’s protocol. Briefly, the protein was extracted from Caco-2 cells using RIPA lysis buffer (Solarbio, Beijing, China). Moreover, the BCA protein assay kit (Beyotime, Shanghai, China) was used to detect the protein concentration. Then, the automated capillary electrophoresis Western System (protein Simple, San Jose, CA, USA) was used to complete all post-sample separations including sample loading, fractionation, immunoprobing, and immunodetection. The calculation and analysis of protein expression were performed according to previous studies [50,51,52]. Meanwhile, primary antibodies of the following proteins were used: glyceraldehyde-3-phosphate dehydrogenase (GAPDH, Cat# ab181602, abcam, 1:100), zonula occludens protein 1(ZO-1, Cat# ab276131, abcam, 1:50), occludin (Cat# ab216327, abcam, 1:50), claudin (Cat# ab180158, abcam, 1:50), Toll-like receptor 4 (TLR4, Cat# ab13556, abcam, 1:25), and p65 (Cat# 4764S, CST, 1:25).

### 4.11. Immunofluorescence Staining

Caco-2 cells were cultured in 12-well plates to assay the relative intensity of ZO-1, occluding, TLR4, and NF-κB p65 through immunofluorescence. After the corresponding treatments, the cells were washed with sterile PBS three times for 5 min and fixed with 4% stationary liquid at 4 °C for 10 min, then washed with sterile PBS three times for 5 min. Then, 600 µL/well of PBST (0.5% Triton X-100) was added for 10 min, washed once with sterile PBS for 5 min, and blocked with 10% goat serum for 1 h at room temperature. Cells were incubated with ZO-1(1:500), Occludin (1:200), TLR4 (1:100), and NF- kB p65 (1:100) overnight at 4 °C. The cells were washed with PBST three times for 5 min. Then, 400 μL/well of Alexa Fluor 488-conjugated goat anti-rabbit IgG secondary antibody (Cat# ab150077, abcam, 1:500) was added for 1 h at 37 °C in the dark and washed with PBST three times for 5 min. Finally, 400 μL/well of Hoechst 33258 was added to counterstain cell nuclei for 5 min. The cells were imaged with a laser scanning confocal microscope (ZEISS, LSM800, Oberkochen, Germany), and the relative fluorescence intensity was acquired using Image J (V1.8.0, National institutes of health, Bethesda, USA).

### 4.12. Statistical Analysis

All statistical analyses were performed by the one-way analysis of variance (ANOVA) using the SPSS 19.0 (SPSS Inc., Chicago, IL, USA). Duncan’s multiple ranges were performed when the data did not follow a normal distribution. Graphpad Prism 8 (GraphPad Software Inc., La Jolla, CA, USA) was used to detect differences among the groups. All data were presented as mean ± SEM, and *p* < 0.05 was considered to be statistically significant.

## 5. Conclusions

In conclusion, the treatment of LPS-induced Caco-2 cells with the *L. rhamnosus* CY12 strain effectively relieved the oxidative stress and improved the intestinal TJ barrier function by increasing the levels of antioxidant enzymes such as CAT, SOD, and GSH-Px and reducing inflammatory cytokine levels, apoptosis, and the destruction of TJ proteins (claudin, occludin, and ZO-1). Different concentrations of the *L. rhamnosus* CY12 strain relieved the degree of inflammatory injury stimulated by LPS; in particular, the concentration of 10^8^ cfu/mL significantly prevented the inflammatory injury induced by LPS in Caco-2 cells. The mechanism was that *L. rhamnosus* CY12 prevented damage to the intestinal epithelial function in LPS-induced Caco-2 cells by upregulating TJ protein expression and reducing ROS accumulation, as well as overproducing inflammatory cytokines that activated the TLR4/NF-κB signaling pathway. Therefore, the *L. rhamnosus* CY12 strain as a novel strain isolated from cattle-yak milk possesses potential effects on the maintenance of overall health and should be considered as a dietary supplementation for food and a promising approach to treat intestinal barrier dysfunction or promote the intestinal TJ barrier function.

## Figures and Tables

**Figure 1 ijms-23-11162-f001:**
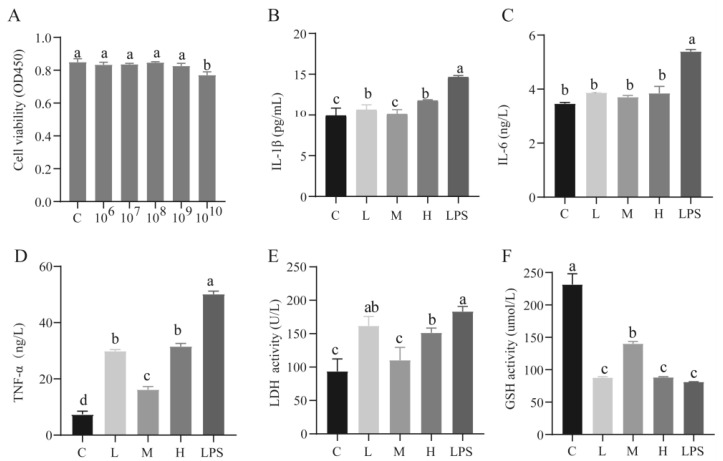
Effects of the pretreatment of different concentration of *L. rhamnosus* CY12 on cell viability, cytokines expression, LDH and GSH activity in LPS-induced Caco-2 Cells. (**A**). Cell viability. (**B**). IL-1β level. (**C**) IL-6 level. (**D**). TNF-α level. (**E**). LDH activity. (**F**). GSH activity. Note: C represents uninduced Caco-2 cells; L, M, and H represent LPS-induced Caco-2 cells co-incubated with 10^7^, 10^8^ and 10^9^ cfu/mL *L. rhamnosus* CY12, respectively; LPS represents LPS-induced Caco-2 cells. Data were presented as mean ± SEM. Means with the different letters are significantly different *p* < 0.05.

**Figure 2 ijms-23-11162-f002:**
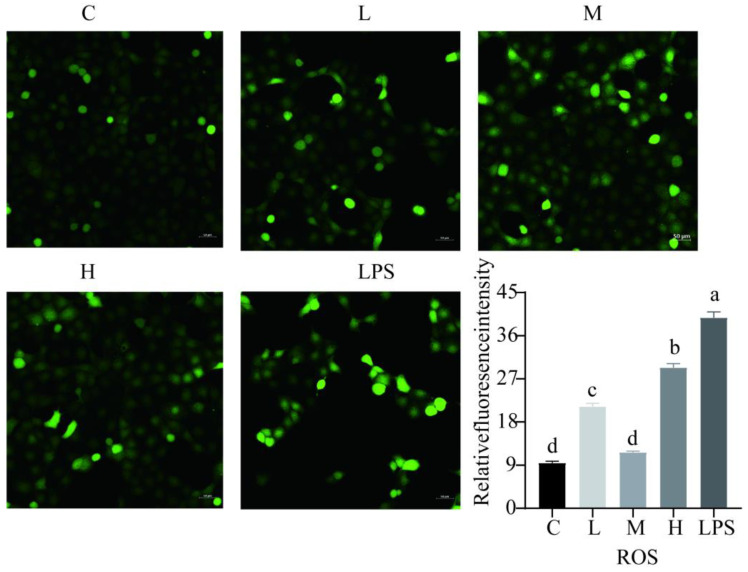
Effects of *L. rhamnosus* CY12 on the intracellular concentrations of ROS levels in LPS-induced Caco-2 Cells. Note: C represents uninduced Caco-2 cells; L, M, and H represent LPS-induced Caco-2 cells co-incubated with 10^7^, 10^8^, and 10^9^ cfu/mL *L. rhamnosus* CY12, respectively; LPS represents LPS-induced Caco-2 cells. Data were presented as mean ± SEM. Means with the different letters are significantly different *p* < 0.05. The magnification of these picture is 100X, and scale bar is 50 µm. Intracellular ROS showed green fluorescent light in the microscope.

**Figure 3 ijms-23-11162-f003:**
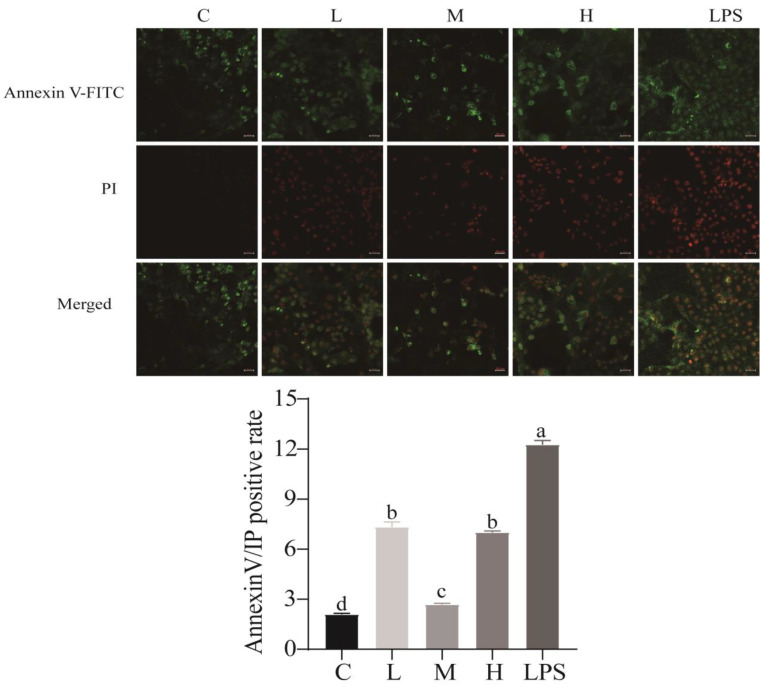
Effects of *L. rhamnosus* CY12 on the intracellular concentrations of apoptosis in LPS-induced Caco-2 Cells. Note: C represents uninduced Caco-2 cells; L, M, and H represent LPS-induced Caco-2 cells co-incubated with 10^7^, 10^8^, and 10^9^ cfu/mL *L. rhamnosus* CY12, respectively; LPS represents LPS-induced Caco-2 cells. Means with the different letters (annexin V/IP positive rate) are significantly different *p* < 0.05. The magnification of these picture is 100X, and scale bar is 50 µm. The early apoptotic cells showed green fluorescent light under the microscope. The late apoptotic cells showed red and green fluorescent light under the microscope, and normal cells displayed no fluorescence.

**Figure 4 ijms-23-11162-f004:**
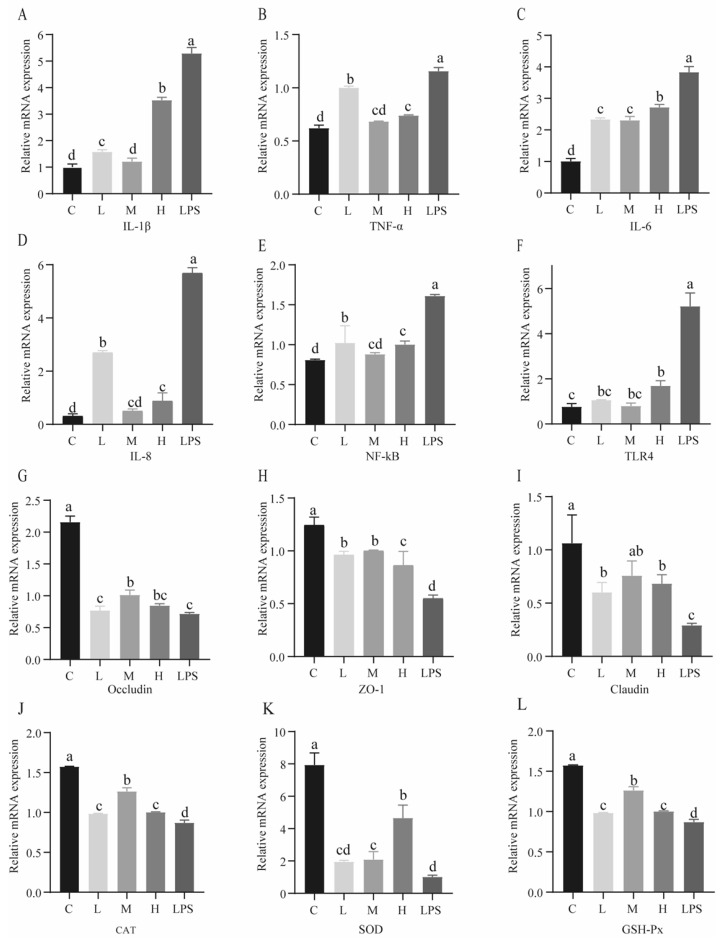
Effects of *L. rhamnosus* CY12 on mRNA expressions related to inflammation, TJ proteins, and antioxidant enzymes in LPS-induced Caco-2 Cells. (**A**): mRNA expression of IL-1β. (**B**): mRNA expression of TNF-α. (**C**): mRNA expression of IL-6. (**D**): mRNA expression of IL-8. (**E**): mRNA expression of NF-κB. (**F**): mRNA expression of TLR4. (**G**): mRNA expression of occludin. (**H**): mRNA expression of ZO-1. (**I**): mRNA expression of claudin. (**J**): mRNA expression of CAT. (**K**): mRNA expression of SOD. (**L**): mRNA expression of GSH-Px. Note: C represents uninduced Caco-2 cells; L, M, and H represent LPS-induced Caco-2 cells co-incubated with 10^7^, 10^8^, and 10^9^ cfu/mL *L. rhamnosus* CY12, respectively; LPS represents LPS-induced Caco-2 cells. Data were presented as mean ± SEM. Means with the different letters are significantly different *p* < 0.05.

**Figure 5 ijms-23-11162-f005:**
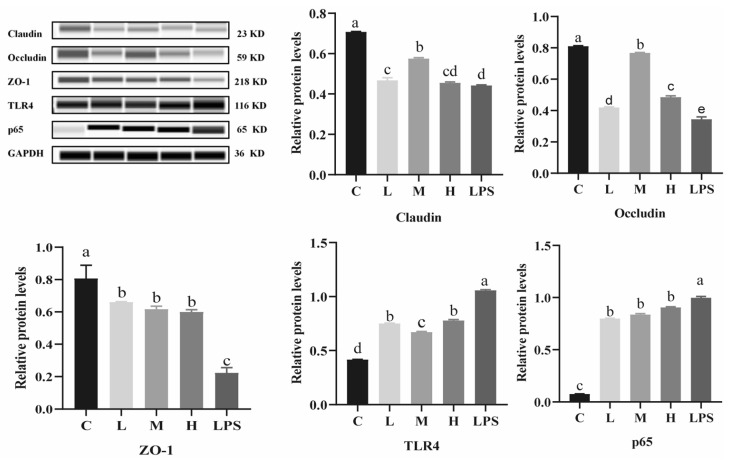
Effects of *L. rhamnosus* CY12 on tight junction and TLR4/p65 protein expression in LPS-induced Caco-2 Cells. Note: C represents uninduced Caco-2 cells; L, M, and H represent LPS-induced Caco-2 cells co-incubated with 10^7^, 10^8^, and 10^9^ cfu/mL *L. rhamnosus* CY12, respectively; LPS represents LPS-induced Caco-2 cells. Data were presented as mean ± SEM. Means with the different letters are significantly different *p* < 0.05.

**Figure 6 ijms-23-11162-f006:**
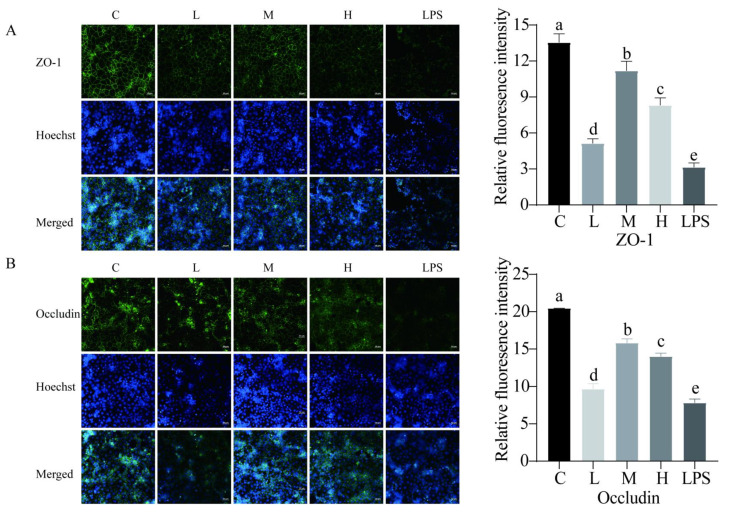
Effects of *L. rhamnosus* CY12 on immunolocalization and the relative fluorescence intensity of tight junction proteins in LPS-induced Caco-2 Cells. (**A**). ZO-1. (**B**). Occludin. Note: C represents uninduced Caco-2 cells; L, M, and H represent LPS-induced Caco-2 cells co-incubated with 10^7^, 10^8^ and 10^9^ cfu/mL *L. rhamnosus* CY12, respectively; LPS represents LPS-induced Caco-2 cells. Data were presented as mean ± SEM. Means with the different letters are significantly different *p* < 0.05. The magnification of these picture is 100X, and scale bar is 50 µm.

**Figure 7 ijms-23-11162-f007:**
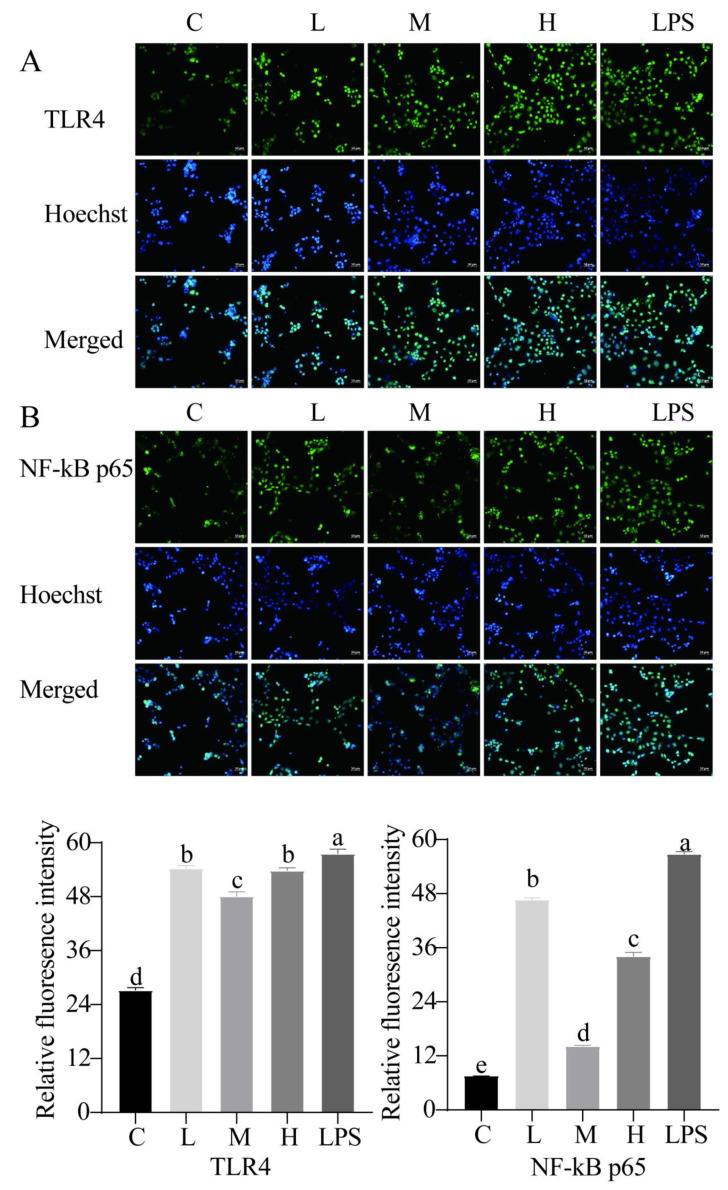
Effects of *L. rhamnosus* CY12 on the immunolocalization and relative fluorescence intensity of TLR4 and NF-κB p65 in LPS-induced Caco-2 Cells. (**A**). TLR4. (**B**). NF-κB p65. Note: C represents uninduced Caco-2 cells; L, M, and H represent LPS-induced Caco-2 cells co-incubated with 10^7^, 10^8^ and 10^9^ cfu/mL *L. rhamnosus* CY12, respectively; LPS represents LPS-induced Caco-2 cells. Data were presented as mean ± SEM. Means with the different letters are significantly different *p* < 0.05. The magnification of these picture is 100X, and scale bar is 50 µm.

**Figure 8 ijms-23-11162-f008:**
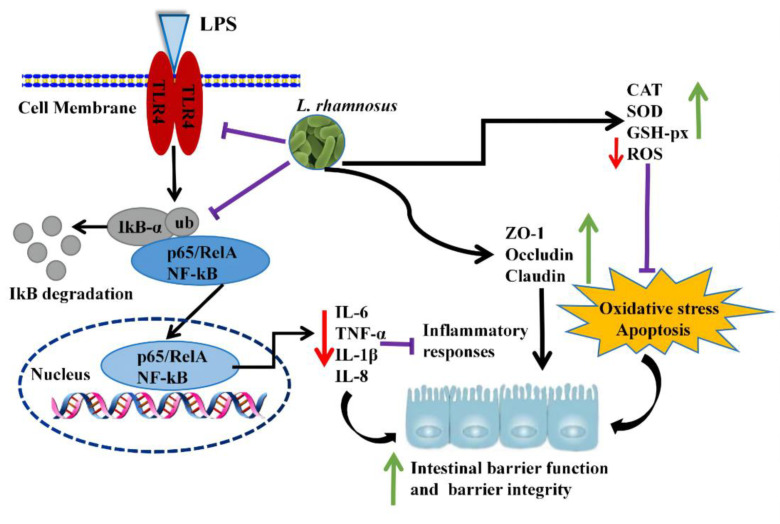
*L. rhamnosus* CY12 prevented LPS-induced intestinal barrier function damage in Caco-2 cells by increasing tight junction protein expression and decreasing oxidative stress and inflammation. Note: green ↑: upregulation of genes or proteins expression; red ↓: downregulation of genes or proteins expression; all black →: promote or result in; purple ⊥: the genes or proteins expression were inhibited.

## Data Availability

Not applicable.

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
