# Peer review of "Lactobacillus rhamnosus CY12 Enhances Intestinal Barrier Function by Regulating Tight Junction Protein Expression, Oxidative Stress, and Inflammation Response in Lipopolysaccharide-Induced Caco-2 Cells"

_ijms, 2022, doi:10.3390/ijms231911162_

Round 1
Reviewer 1 Report
Peer Review Repor:
Original Submission:
1. Recommendation: Minor Revision
2. Comments to authors
Overview and general recommendation: Overall, the study is well designed, the experiments are well performed and explained. The results are clearly presented and discussed. Although the flaws within the manuscript, I suggest its publication in case of minor revision. Some indications for minor revisions are given below.
line 32: of TJ protein. Try to change it to 'TJ proteins', all through the text because we have three different proteins, not only one.
line 34: Lactobacillus rhamnosus CY12 as probiotics. Change it as follows "Lactobacillus rhamnosus CY12 as a probiotic"
line 55: " oxidative stress appears...". Change it to following form: " oxidative stress seems to...". The meaning is more adequate.
line 58: "in vitro". The italic mode.
line 61: "...E. coli..". Put the bacterial name in italics.
line 73: "the application of the lactic acid bacteria is considered to be an important strategy...". More precision is needed. Change it as follows: "the application of beneficial microbes also known as 'probiotics' is considered to be an important strategy...".
line 78: "the Bifidobacterium bifidum (B. bifidum) strain". Please put the italic mode for the bacterial name all through the text.
Line 83: The italic mode!
line 95: "to be studied furthIn...". Rectify the sentence to make the sense clear. "o be studied further. In the present study, ...".
line 162: "AND", there is no need to use capital letters.
line 230: (Figure 6): the right protein is Occludin and not Claudin in (B)!
line 236: (Figure 7): (A). ZO-1. False, you have to change it to (A). TLR4
line 241 to line 243: Reformulate the whole passage. The meaning is not clear.
line 246: "and provoked chronic inflammatory diseases.". Try to add a reference for all the aforementioned information.
line 250: "...30], Our results...". It should be written like this. "...30], our results...".
line 254 to line 256: Reformulate it. The sense is ambiguous.
line 259: "...tight junction protein expressions,". Change it as follows: "...tight junction proteins expression,".
line 268: "the potential use of the L. rhamnosus CY12 strain as probiotics.". Change it as follows: "the potential use of the L. rhamnosus CY12 strain as a promising probiotic candidate."
line 274: "...played important role...". Change it as follows: "... played an important role...".
line 276: Change the term "balance" to the term "homeostasis". "...the intestinal homeostasis [26]."
line 279, line 438: "..TJ protein expressions...". Change it as follows: "...TJ proteins expression..."
line 285: "...the most representative of which is IL-6,...". You have to change it to the following form: "...the major representatives of which areIL-6, ...". It is more adequate.
line 290: "... similar to our results...". Try to change it as follows: "... which are in line with our findings."
line 316: "...obtained in pellet, the pellet was washed...". Change it as follows: "...obtained in pellet. This latter was washed...". It is more suitable.
line 381, line 394, line 410: "The 6-cell plates..". You mean the 6-well plates!
line 413: "...then wished...". You mean "washed"!
Try to develop the introduction with more recent and relevant references in the field of probiotics and beneficial microbes such as Vijayaram and Kannan (2018); Zommiti et al. 2020; etc
Try to develop the conclusion,
Are there any horizons of potential use of this strain at an animal, human or industrial scale?
Check all the text in order to put commas and punctuation in right places to give the meaning to the sentences and ideas.
Check all the text for italic mode.
Verify the english language and refine the writing style.
In terms of experiments, they are well performed and clearly presented.
Thank you for all the work done!
Author Response
Reviewer #1
Comment 1
line 32: of TJ protein. Try to change it to 'TJ proteins', all through the text because we have three different proteins, not only one.
line 34: Lactobacillus rhamnosus CY12 as probiotics. Change it as follows "Lactobacillus rhamnosus CY12 as a probiotic"
line 55: " oxidative stress appears...". Change it to following form: " oxidative stress seems to...". The meaning is more adequate.
line 58: "in vitro". The italic mode.
line 61: "...E. coli..". Put the bacterial name in italics.
line 73: "the application of the lactic acid bacteria is considered to be an important strategy...". More precision is needed. Change it as follows: "the application of beneficial microbes also known as 'probiotics' is considered to be an important strategy...".
line 78: "the Bifidobacterium bifidum (B. bifidum) strain". Please put the italic mode for the bacterial name all through the text.
Line 83: The italic mode!
line 95: "to be studied furthIn...". Rectify the sentence to make the sense clear. "o be studied further. In the present study, ...".
line 162: "AND", there is no need to use capital letters.
line 230: (Figure 6): the right protein is Occludin and not Claudin in (B)!
line 236: (Figure 7): (A). ZO-1. False, you have to change it to (A). TLR4
line 241 to line 243: Reformulate the whole passage. The meaning is not clear.
line 246: "and provoked chronic inflammatory diseases.". Try to add a reference for all the aforementioned information.
line 250: "...30], Our results...". It should be written like this. "...30], our results...".
line 254 to line 256: Reformulate it. The sense is ambiguous.
line 259: "...tight junction protein expressions,". Change it as follows: "...tight junction proteins expression,".
line 268: "the potential use of the L. rhamnosus CY12 strain as probiotics.". Change it as follows: "the potential use of the L. rhamnosus CY12 strain as a promising probiotic candidate."
line 274: "...played important role...". Change it as follows: "... played an important role...".
line 276: Change the term "balance" to the term "homeostasis". "...the intestinal homeostasis [26]."
line 279, line 438: "..TJ protein expressions...". Change it as follows: "...TJ proteins expression..."
line 285: "...the most representative of which is IL-6,...". You have to change it to the following form: "...the major representatives of which areIL-6, ...". It is more adequate.
line 290: "... similar to our results...". Try to change it as follows: "... which are in line with our findings."
line 316: "...obtained in pellet, the pellet was washed...". Change it as follows: "...obtained in pellet. This latter was washed...". It is more suitable.
line 381, line 394, line 410: "The 6-cell plates..". You mean the 6-well plates!
line 413: "...then wished...". You mean "washed"!
Response 1
Thanks for your suggestions, we have revised and been consistent in whole paper.
Comment 2
Try to develop the introduction with more recent and relevant references in the field of probiotics and beneficial microbes such as Vijayaram and Kannan (2018); Zommiti et al. 2020; etc
Response 2
The reviewer is right. In doing so, we have tried to the introduction with more recent and relevant references in the field of probiotics and beneficial microbes. See for details: “Additionally, potential probiotic microorganisms and probiotics have demonstrated an astonishing antagonistic activity toward a wide range of sturdy food and clinical pathogens[24, 25]. The importance of probiotics in human-animal nutrition is widely recognized[26].”As page 2, lines 77-80.
Comment 3
Try to develop the conclusion
Response 3
We have developed the conclusion. See for details:“Therefore, L. rhamnosus CY12 strain as a novel strain isolated from cattle-yak milk possesses potential effects on the maintenance of overall health and should be considered as a dietary supplementation for food and a promising approach to treat intestinal barrier dysfunction or promote the intestinal TJ barrier function.” Lines 452-456.
Comment 4
Are there any horizons of potential use of this strain at an animal, human or industrial scale?
Response 4
Thanks for your suggestions and question, L. rhamnosus CY12 strain as a novel strain isolated from cattle-yak milk, our previous work reported L. rhamnosus CY12 strain displayed potential probiotic characteristics such as high survival rate in acidic condition and bile salts, high antimicrobial activity and adhesive potential. In addition, this study indicated that L. rhamnosus CY12 could relieve cytotoxicity, apoptosis, oxidative stress, and pro-inflammatory cytokine expressions, and also inhibit the TLR4/NF-κB signaling pathway. Furthermore, the gene expression of antioxidant enzymes, as well as the mRNA and protein expressions of TJ proteins was improved. Particularly the concentration of 108 cfu/mL significantly prevented the inflammatory injury induced by LPS in Caco-2 cells. Therefore, L. rhamnosus CY12 as a novel strain isolated from cattle-yak milk possesses value as a potential probiotic or dietary supplementation for food, for treating intestinal disorders.
Comment 5
Check all the text in order to put commas and punctuation in right places to give the meaning to the sentences and ideas.
Response 5
Thanks for your suggestions, we have revised in whole paper.
Comment 6
Check all the text for italic mode.
Response 6
Corrected as suggested.
Comment 7
Verify the english language and refine the writing style.
Response 7
We appreciate the feedback from the reviewer. According your suggestions, we tried our best to alter all grammatical mistakes throughout the manuscript. We have revised the whole paper, the academic and professional quality of the article has been improved.

Reviewer 2 Report
I would like to express my appreciation for the subject matter of the manuscript. The search for natural substances that can be a source of valuable substances for medicine is a very important and useful research task. Therefore, such publications are of great value.
However, I believe that your work needs to be improved, at least with a better, more detailed discussion of the results also needs other improvements:
1. Kindly I suggest to concise he background section of the abstract part to be clear for readers.
2. After mentioning the full name of Lactobacillus rhamnosus CY12, you then can use abbreviated manner as L. rhamnosus CY12.
3. Add briefly the used techniques for the evaluation of the effects of L. rhamnosus CY12 on oxidative stress, inflammation, and disruption of tight junction.
4. For all your experiments, add negative and positive controls to validate the used methods.
5. In the methods section add suitable references (updated) to all the methods section.
6. Expand the conclusion part to contain your personal recommendations for your outcomes and your work future plan in the supplements industry.
7. Add clearer graphs for figures 3 & 7.
8. The whole manuscript needs major grammar, typo and editing corrections
Author Response
Reviewer #2:
Comment 1
Kindly I suggest to concise the background section of the abstract part to be clear for readers.
Response 1
Thank you for your suggestions. The revised version as follows: “The intestinal barrier is vital for preventing inflammatory bowel disease (IBD). The objectives of this study were to assess whether the Lactobacillus rhamnosus CY12 could alleviate oxidative stress, inflammation, and disruption of tight junction (TJ) barrier function induced by lipopolysaccharide (LPS), and therefore to explore the potential underlying molecular mechanisms. ” Lines 20-24.
Comment 2
After mentioning the full name of Lactobacillus rhamnosus CY12, you then can use abbreviated manner as L. rhamnosus CY12
Response 2
Done it as your suggestion.
Comment 3
Add briefly the used techniques for the evaluation of the effects of L. rhamnosus CY12 on oxidative stress, inflammation, and disruption of tight junction
Response 3
Thank you very much for your valuable advice. The oxidative stress levels were evaluated through LDH activity, GSH activity, ROS levels with corresponding assay kits, and the mRNA expressions of antioxidant enzymes in each group using qPCR technique. The inflammatory response was evaluated with the ELISA, qPCR, and the capillary western blot, and immunofluorescence techniques. The tight junction was evaluated with qPCR, immunofluorescence, and the capillary western blot techniques. Detailed descriptions were provided in Materials and Methods 4.4,4.5,4.6,4.7,4.9,4.10,and 4.11.
Comment 4
For all your experiments, add negative and positive controls to validate the used methods
Response 4
Thank you for your suggestions, we apologized for the lack of clarity in the description of our methods. In our methods, the control group without L. rhamnosus CY12 or LPS for negative control group; the low to high dose groups were treated with 107 cfu/mL L. rhamnosus CY12 and 1 μg/ml LPS, 108 cfu/mL L. rhamnosus CY12 and 1 μg/ml LPS, and 109 cfu/mL L. rhamnosus CY12 and 1 μg/ml LPS, respectively; and the LPS group was treated with 1 μg/ml LPS for positive control group. See for details: “The Caco-2 cells were treated with the following: (1) the control group (C): without L. rhamnosus CY12 or LPS for negative control group; (2) Low-dose group (L): treated with 107 cfu/mL L. rhamnosus CY12 and 1 μg/ml LPS; (3) Middle-dose group (M): treated with 108 cfu/mL L. rhamnosus CY12 and 1 μg/ml LPS; (4) High-dose group (H): treated with 109 cfu/mL L. rhamnosus CY12 and 1 μg/ml LPS; (5) LPS group (LPS): 1 μg/ml LPS induced Caco-2 cells for positive control group.” Lines 348-353.
Comment 5
In the methods section add suitable references (updated) to all the methods section
Response 5
Done suggested.
Comment 6
Expand the conclusion part to contain your personal recommendations for your outcomes and your work future plan in the supplements industry.
Response 6
Thank you for your comment, we have developed the conclusion. See for details:“Therefore, L. rhamnosus CY12 strain as a novel strain isolated from cattle-yak milk possesses potential effects on the maintenance of overall health and should be considered as a dietary supplementation for food and a promising approach to treat intestinal barrier dysfunction or promote the intestinal TJ barrier function.” Lines 452-456.
Comment 7
dd clearer graphs for figures 3 & 7.
Response 7
Done suggested.
Comment 8
The whole manuscript needs major grammar, typo and editing corrections
Response 8
We appreciate the feedback from the reviewer. According your suggestions, we tried our best to alter all grammatical mistakes throughout the manuscript. We have revised the whole paper, the academic and professional quality of the article has been improved.

Round 2
Reviewer 2 Report
The authors established all the required corrections